# Effects of Reducing L-Carnitine Supplementation on Carnitine Kinetics and Cardiac Function in Hemodialysis Patients: A Multicenter, Single-Blind, Placebo-Controlled, Randomized Clinical Trial

**DOI:** 10.3390/nu13061900

**Published:** 2021-05-31

**Authors:** Miki Sugiyama, Takuma Hazama, Kaoru Nakano, Kengo Urae, Tomofumi Moriyama, Takuya Ariyoshi, Yuka Kurokawa, Goh Kodama, Yoshifumi Wada, Junko Yano, Yoshihiko Otsubo, Ryuji Iwatani, Yukie Kinoshita, Yusuke Kaida, Makoto Nasu, Ryo Shibata, Kyoko Tashiro, Kei Fukami

**Affiliations:** 1Division of Nephrology, Department of Medicine, Kurume University School of Medicine, Kurume, Fukuoka 830-0011, Japan; sugiyama_miki@med.kurume-u.ac.jp (M.S.); hazama_takuma@kurume-u.ac.jp (T.H.); nakano_kaoru@med.kurume-u.ac.jp (K.N.); urae_kengo@med.kurume-u.ac.jp (K.U.); moriyama_tomofumi@med.kurume-u.ac.jp (T.M.); ariyoshi_takuya@med.kurume-u.ac.jp (T.A.); kurokawa_yuka@med.kurume-u.ac.jp (Y.K.); kodama_gou@med.kurume-u.ac.jp (G.K.); akkun@med.kurume-u.ac.jp (J.Y.); kaida_yuusuke@kurume-u.ac.jp (Y.K.); nasusuma0210@med.kurume-u.ac.jp (M.N.); ryo0513@med.kurume-u.ac.jp (R.S.); 2Sugi Hospital, Omuta, Fukuoka 837-0916, Japan; 3Wada Cardiovascular Clinic, Tosu, Saga 841-0071, Japan; houseikaiwada@outlook.jp; 4Kurume Ekimae Clinic, Kurume, Fukuoka 830-0023, Japan; 5Shin Koga Clinic, Kurume, Fukuoka 830-0033, Japan; y.otsubo@tenjinkai.or.jp; 6Koga 21 Hospital, Kurume, Fukuoka 839-0801, Japan; iwatani_ryuuji@kurume-u.ac.jp; 7Research Institute of Medical Mass Spectrometry, Kurume University School of Medicine, Kurume, Fukuoka 830-0011, Japan; kinoshita_yukie@med.kurume-u.ac.jp (Y.K.); k_tashiro23@med.kurume-u.ac.jp (K.T.)

**Keywords:** acylcarnitine, brain natriuretic peptide, cardiac function, cardiomyopathy, carnitine deficiency, CPT2, end-stage kidney disease, free fatty acid, heart failure, hemodialysis

## Abstract

L-carnitine (LC) supplementation improves cardiac function in hemodialysis (HD) patients. However, whether reducing LC supplementation affects carnitine kinetics and cardiac function in HD patients treated with LC remains unclear. Fifty-nine HD patients previously treated with intravenous LC 1000 mg per HD session (three times weekly) were allocated to three groups: LC injection three times weekly, once weekly, and placebo, and prospectively followed up for six months. Carnitine fractions were assessed by enzyme cycling methods. Plasma and red blood cell (RBC) acylcarnitines were profiled using tandem mass spectrometry. Cardiac function was evaluated using echocardiography and plasma B-type natriuretic peptide (BNP) levels. Reducing LC administration to once weekly significantly decreased plasma carnitine fractions and RBC-free carnitine levels during the study period, which were further decreased in the placebo group (*p* < 0.001). Plasma BNP levels were significantly elevated in the placebo group (*p* = 0.03). Furthermore, changes in RBC (C16 + C18:1)/C2 acylcarnitine ratio were positively correlated with changes in plasma BNP levels (β = 0.389, *p* = 0.005). Reducing LC administration for six months significantly decreased both plasma and RBC carnitine levels, while the full termination of LC increased plasma BNP levels; however, it did not influence cardiac function in HD patients.

## 1. Introduction

Heart failure (HF), as well as some of its complications, such as pulmonary edema, is a serious condition characterized by decreased myocardial contractility and abnormal hemodynamic state in patients with end-stage kidney disease (ESKD). The United States Renal Data System revealed that an estimated 44% of patients on hemodialysis (HD) have chronic HF (CHF), and 5.4% of HD patients die of chronic HF [1] (https://render.usrds.org/2016/view/v2_09.aspx. Accessed date: 8 May 2021). The Japanese Society of Dialysis Transplantation has reported that HF is the most common cause of death in patients undergoing dialysis [2]. Furthermore, CHF is associated with impairments in activities of daily living and quality of life in HD patients [3]. Taken together, the prevention of HF development is a crucial therapeutic strategy for HD patients worldwide.

Carnitine is a natural substance that plays an important role in fatty acid β-oxidation and energy production in mitochondria [4]. Organic cation/carnitine transporter 2 (OCTN2) is capable of transporting free carnitine into the cytoplasm. In addition, carnitine palmitoyltransferase 1 (CPT1) and CPT2 are the mitochondrial outer and inner membrane enzymes, respectively. These enzymes are responsible for delivering long-chain fatty acids into mitochondria, leading to β-oxidation and ATP synthesis. A lack of carnitine and dysfunction of OCTN2 and/or CPT2 results in the inability to produce energy from long-chain fatty acids, leading to the development of cardiomyopathy [5,6]. We, along with others, reported that serum carnitine levels are significantly decreased in HD patients due to the elimination of serum carnitine from the blood via HD [7,8]. Accordingly, HD-related carnitine deficiency may be one of the causative factors for the progression of HF in patients with ESKD. A meta-analysis demonstrated that L-carnitine (LC) supplementation may improve clinical symptoms and cardiac function and decrease serum levels of B-type natriuretic peptide (BNP) and NT-proBNP in patients with CHF [9]. Furthermore, LC supplementation for a year might improve cardiac function in HD patients [10], suggesting that LC treatment may have protective effects on HF in HD patients with carnitine deficiency.

In a recent study, we observed that plasma carnitine concentration is approximately ten-fold higher in HD patients treated with intravenous LC administration at a dose of 1000 mg three times weekly than in those receiving no LC treatment. This finding could be related to the fact that circulating carnitine cannot be excreted in urine and can only be eliminated by dialysis [11]. Furthermore, six months of intravenous LC treatment at a dose of 2000 mg per HD session increased muscle carnitine concentration by three-fold in HD patients [12]. However, the kinetics of serum carnitine fractions and changes in cardiac function after reducing or stopping LC treatment in HD patients remain unknown. Therefore, we prospectively examined the effects of reducing or stopping LC therapy on carnitine concentrations in plasma and red blood cells (RBCs) and cardiac function in HD patients treated with LC.

## 2. Materials and Methods

### 2.1. Patients and Study Protocol

This multicenter, single-blind, placebo-controlled, randomized clinical trial was conducted at Kurume University Hospital, Sugi Hospital, Wada Cardiovascular Clinic, Kurume Ekimae Clinic, Shin Koga Clinic, and Koga 21 Hospital. We recruited a total of 64 HD patients from December 2018 to June 2020. Patients over 20 years of age with ESKD undergoing HD who could provide written informed consent for study participation were enrolled in this study. All patients had already been diagnosed with dialysis-associated secondary carnitine deficiency and had been administered intravenous LC at a dose of 1000 mg per HD session three times weekly for at least three months. The exclusion criteria were as follows: under 20 years of age, unstable lower limb cramps and general fatigue, hemoglobin (Hb) levels below 10 g/dL, and patients deemed to have inadequate information by a physician.

Six months after the registration period, seven patients were excluded before randomization. Among these seven patients, one had Hb levels below 10 g/dL, one did not receive LC, one was referred to another clinic, and four had been administered LC once weekly (Figure 1). The remaining 57 patients were finally included and randomly assigned using simple randomization procedures (computer-generated list of random numbers) by the clinical research coordinator (CRC). They were allocated to LC 1000 mg (5 mL) three times weekly (LC-3) (*n* = 19), LC 1000 mg once and placebo (saline 5 mL) twice weekly (LC-1) (*n* = 18), or placebo three times weekly (LC-0) (*n* = 20) groups (Figure 1).

The participants were blinded to the intervention after the assignment. At baseline, 30 days, 90 days, and 176 days of the study period, patients provided a complete history and underwent physical examination, including blood pressure (BP) and blood chemistry test, just before HD sessions at two-day intervals. The study protocol is shown in Figure 2. All data and samples were collected by the CRC at Kurume University Hospital. The patients were dialyzed for 4–5 h with high-flux dialyzers three times weekly. All patients received 1000 mg of LC intravenously immediately after HD sessions. The primary endpoint was the effects of dose reduction or discontinuation of LC on the kinetics of plasma and RBC carnitine concentration and cardiac function. Additional analyses were performed to evaluate changes in plasma BNP levels before and after the study period.

### 2.2. Clinical, Demographic, and Anthropometric Measurements

The patients’ medical histories were ascertained using a questionnaire. Vigorous physical activity and smoking were avoided for at least 30 min before HD sessions. Blood was drawn from an arteriovenous shunt just before starting the HD sessions to determine Hb, total protein, serum albumin, total cholesterol, low-density lipoprotein cholesterol, blood urea nitrogen, calcium, and phosphate levels. These parameters were analyzed by commercially available laboratories (Daiichi Pure Chemicals, Tokyo, Japan, and Wako Pure Chemical Industries, Osaka, Japan). Plasma BNP was measured using a chemiluminescent immunoassay (CRC Corporation, Fukuoka, Japan). Plasma carnitine fraction (total carnitine, free carnitine, and acylcarnitine) levels were determined by an enzymatic cycling method, as previously described [13]. Free carnitine and acylcarnitines in plasma and RBC were profiled using tandem mass spectrometry at baseline, 30 days, 90 days, and 176 days after beginning the study, according to a previously described method [14,15]. Changes in plasma BNP and acylcarnitine ratio were defined as the differences between the baseline values (pre-data) and those at 176 days (post-data). Changes in these data were calculated using the following formula: (post-data–pre-data)/pre-data × 100 (%).

### 2.3. Evaluation of Cardiac Function

Systolic and diastolic cardiac functions were evaluated using transthoracic echocardiography by professional technicians at each hospital who were unaware of the participants’ clinical data. Left ventricular (LV) mass index (LVMI) was determined using the standard formula as follows: Devereux formula = 1.04 × ((LV end-diastolic diameter + interventricular septal thickness at end-diastole + posterior wall thickness at end-diastole)^3^ − (LV end-diastolic diameter)^3^) − 13.6 [16]. LV ejection fraction (LVEF) was calculated to determine systolic myocardial function using the standard biplane Simpson’s method from the formula: LVEF = ((end-diastolic volume) − (end-systolic volume))/end-diastolic volume. LV filling index (E/e’) was calculated as the ratio of the transmitral flow velocity to the annular velocity as a marker of diastolic cardiac function.

### 2.4. Statistical Analysis

To compare clinical variables at baseline and on day 176 of the study period, the paired t-test was used. Non-parametric analysis (Wilcoxon signed-rank test) was used to determine differences between baseline plasma and RBC-free carnitine because the data did not follow a normal distribution using the Shapiro–Wilk test. One-way ANOVA with post-hoc Tukey HSD test was used to compare the data among the three groups. For exploratory data analysis, univariate and multiple regression analyses were performed to determine correlations between changes in BNP levels and clinical variables, including acylcarnitine ratios. Natural logarithmic transformation was used due to the skewness of the BNP distribution. Data are presented as mean ± standard deviation. Statistical significance was set at *p* < 0.05, and all statistical analyses were performed using the JMP Pro version.15 software (SAS Institute Inc., Cary, NC, USA).

## 3. Results

### 3.1. Demographic Data at Baseline

During the study period, two patients were referred to another clinic, one died due to gastrointestinal bleeding, and one patient discontinued treatment due to restless legs syndrome. Two patients were excluded from the study because RBC acylcarnitine levels were not measured. Finally, a total of 51 patients completed the treatment in the groups: LC-3 group (*n* = 18), LC-1 group (*n* = 16), and LC-0 group (*n* = 17) (Figure 1). No significant adverse events related to LC treatment stoppage were observed during the study period. No significant differences in age, HD duration, duration of previous LC treatment, or other clinical parameters, except for phosphate, were observed among the groups. The baseline phosphate level was higher in the LC-0 group than in the LC-1 group (*p* = 0.038) (Table 1). The baseline plasma total and free carnitines, acylcarnitine, and RBC-free carnitine levels were extremely high in all groups and were not different among the groups. The baseline free carnitine levels in RBCs were significantly higher than those in plasma in patients assessed by tandem mass spectrometry (462 ± 215, 293 ± 66 μmol/L, *p* < 0.0001). The baseline cardiac function and BNP levels were not different among the study groups.

### 3.2. Effects of Reducing or Stopping LC Administration on Clinical Variables and Carnitine Fractions

In Table 2, the comparison of clinical variables and carnitine fractions between baseline and day 176 of the study period is represented by the *p*-value. After the study period, body weight significantly decreased and the total protein, albumin, and total and LDL-cholesterol levels in the serum increased in the LC-3 group rather than the LC-1 and LC-0 groups (Table 2).

Significant decreases in the plasma total carnitine, free carnitine, and acylcarnitine levels assessed by enzyme cycling methods were observed at 30 days after reducing or stopping LC administration. In addition, all carnitine fraction levels were further reduced at 90 and 176 days in the LC-1 and LC-0 groups (*p* < 0.0001) (Figure 3A–C). Significant differences in plasma carnitine fraction levels between the LC-1 and LC-0 groups were similarly observed in this study (Figure 3A–C). The acyl/free carnitine ratio was not affected by the reduction or cessation of LC therapy during the study (Figure 3D).

In addition, we examined free carnitine levels in RBCs, reflecting tissue carnitine levels. Free carnitine levels in RBCs were decreased in the LC-1 and LC-0 groups compared with those in the LC-3 group during the study (*p* < 0.0001) (Figure 4). Similarly, there was a significant difference in free carnitine levels in RBCs between the LC-1 and LC-0 groups during the study (Figure 4).

### 3.3. Effects of Reducing or Stopping LC Administration on Cardiac Function and Plasma BNP Levels

Reducing or stopping LC administration did not change systolic and diastolic cardiac functions assessed by LVEF and E/e’, respectively. There was no difference in LVMI before and after the study period in any of the groups. However, only stopping LC treatment for six months significantly increased BNP levels (*p* = 0.03) (Table 3).

### 3.4. Correlations between Changes in BNP Levels and Clinical Variables and Acylcarnitine Ratios

To further explore which variables could be independently associated with BNP levels, we assessed correlations between changes in BNP and clinical variables and acylcarnitine ratios, such as (C16 + C18:1)/C2, a marker of CPT2 deficiency; C0/(C16 + C18), a marker of CPT1 deficiency; C8/C10, a marker of medium-chain acyl-CoA dehydrogenase deficiency; and C14/C3, another marker of CPT2 deficiency, by univariate and multiple regression analyses in all HD patients. We speculated that independent determinants of the changes in BNP might be involved in the pathogenesis of carnitine deficiency-associated HF. Changes in systolic BP were positively associated with changes in BNP levels (β = 0.373, SE = 0.152, *p* = 0.007) (Table 4). Changes in plasma (C16 + C18:1)/C2, C0/(C16 + C18), C8/C10, and C14/C3 were not associated with changes in BNP levels (Table 4) (Figure 5A). Although changes in RBC C0/(C16 + C18), C8/C10, and C14/C3 were not associated with BNP, changes in RBC (C16 + C18:1)/C2 were positively correlated with changes in BNP levels (β = 0.398, SE = 0.283, *p* = 0.005) (Table 4) (Figure 5B). Both changes in systolic BP and RBC (C16 + C18:1)/C2 were the sole independent determinants of changes in BNP levels in these patients (R^2^ = 0.259) (Table 4).

## 4. Discussion

In this study, we demonstrated that reducing or stopping LC administration for six months significantly decreased all plasma carnitine fractions and RBC-free carnitine levels. In addition, stopping LC treatment significantly increased plasma BNP levels. Moreover, changes in systolic BP and RBC (C16 + C18:1)/C2 values were the sole independent determinants of changes in BNP levels. To the best of our knowledge, this is the first report to demonstrate that stopping LC administration increased BNP levels in HD patients treated with LC.

Translocation of long-chain free fatty acids into mitochondria by carnitine shuttle plays a pivotal role in energy production via β-oxidation [17]. Therefore, carnitine deficiency induces cardiac dysfunction and muscle atrophy in HD patients. We previously reported a significant decrease in plasma carnitine fraction levels in HD patients [8]. Higuchi et al. reported the beneficial effects of LC on cardiac function and LVMI assessed by echocardiography and NT-proBNP in carnitine-deficient HD patients [10]. However, the effects of reducing or stopping LC on cardiac function have not yet been reported. In this study, we demonstrated that reducing or stopping LC administration for six months significantly reduced plasma and RBC carnitine fraction levels. Furthermore, plasma BNP levels significantly increased only in patients who stopped LC treatment, suggesting that stopping LC for six months may contribute to the future progression of cardiac dysfunction in HD patients. Indeed, suspension of LC therapy has been reported to result in signs and symptoms of recurrence (muscular weakness, fatigue, cardiac enlargement, and low cardiac function) in patients with primary carnitine deficiency treated with LC [18]. However, regardless of the increase in BNP levels, LV function assessed by EF and E/e’, markers of systolic and diastolic cardiac function, respectively, and LVMI did not change before or after reduction or discontinuation of LC therapy. This result might be explained by the concentration of carnitine fractions during the study. At the end of the study, free carnitine levels in the plasma were still within the normal range, even after stopping the LC therapy for six months. These findings suggest that the inadequate deficiency of carnitine by stopping LC therapy might not have influenced cardiac function in these patients. Taken together, long-term studies are needed to clarify the deleterious effects of stopping LC administration on cardiac function in carnitine-deficient HD patients.

In our study, changes in systolic BP and RBC (C16 + C18:1)/C2 ratio, a marker of CPT2 deficiency, were only positively correlated with changes in plasma BNP levels. CPT2 deficiency in humans is diagnosed by abnormal acylcarnitine profile obtained from blood spotted on filter paper with an increased (C16 + C18:1)/C2 ratio [19]. CPT2 plays a central role in ATP synthesis in cardiomyocytes mitochondria. Genetic deletion of mouse CPT2 induces cardiac hypertrophic remodeling, which activates the mammalian target of the rapamycin complex, thereby reducing protein acetylation and accumulating lipid droplets [20]. Furthermore, LC supplementation is effective in severe CPT2 deficiency [21], indicating that stopping LC supplementation may influence the activity of CPT2 in cardiomyocytes, leading to an increase in BNP levels in HD patients. Furthermore, we found that changes in the (C16 + C18:1)/C2 ratio in RBC, not in plasma, were associated with changes in plasma BNP levels. Acylcarnitine levels in RBCs are higher than those in plasma [22]. In addition, elevated RBC carnitine levels by LC administration decreased very slowly even after irrigation suggesting a low carnitine turnover in RBCs [22]. These findings indicate that the acylcarnitine ratio in RBC, rather than that in plasma, appears to be an accurate tissue acylcarnitine profile in HD patients.

This study has several limitations. First, the sample size of the patients was small; thus, the statistical power was weak. Second, the short study duration might have affected the efficacy of reducing or stopping LC in cardiac function. Therefore, further large and long-term clinical studies are warranted to verify the effect of reduction in LC treatment on the cardiac function of HD patients.

Carnitine deficiency induces mitochondrial damages through the impairment of β-oxidation and ATP production. However, the exact mechanism underlying the exacerbation of cardiomyocyte injury by hemodialysis-induced carnitine deficiency through the manipulation of carnitine-related mitochondrial enzymes, such as CPT1 and CPT2, is not fully understood. Hence, future research using a carnitine-deficient experimental model might be required to examine the mediating mechanism underlying carnitine deficiency-associated myocardial injury.

## 5. Conclusions

In conclusion, reducing LC administration for six months significantly decreased both plasma and RBC carnitine levels. Moreover, stopping LC increased plasma BNP levels; however, this stoppage did not influence cardiac function in HD patients.

## Figures and Tables

**Figure 1 nutrients-13-01900-f001:**
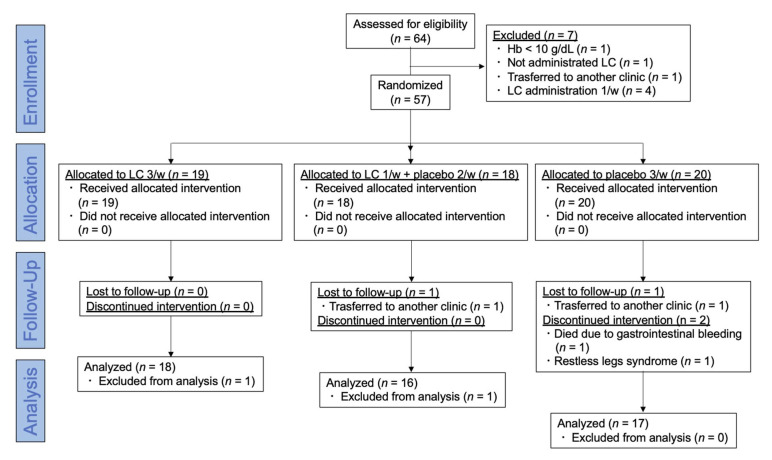
CONSORT flow diagram of this trial. Sixty-four hemodialysis patients were assessed for eligibility, and 57 were randomized and allocated to LC 3/week, LC 1/week + placebo 2/week, and placebo 3/week groups and followed up for six months. Hb: hemoglobin; LC: L-carnitine.

**Figure 2 nutrients-13-01900-f002:**
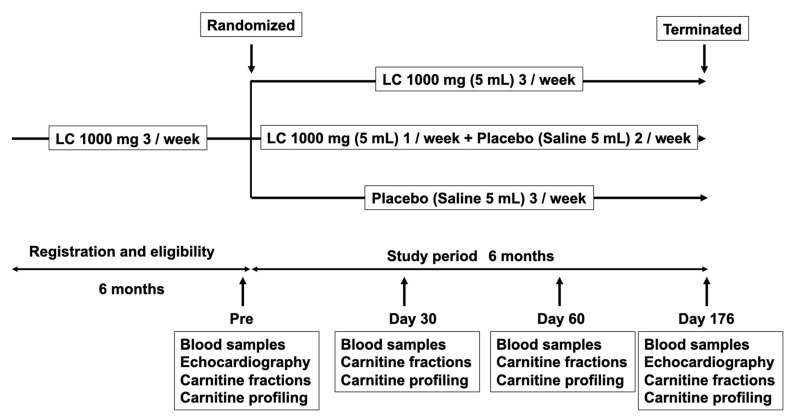
Trial design. Patients were randomized to LC 1000 mg 3/week, LC 1000 mg 1/week + placebo 2/week, and placebo 3/week groups and followed up for six months. Blood samples, including carnitine fractions and profiling, were examined before the study period, day 30, day 60, and day 176 of the study period. Echocardiography was evaluated before and on day 176 of the study period. LC: L-carnitine.

**Figure 3 nutrients-13-01900-f003:**
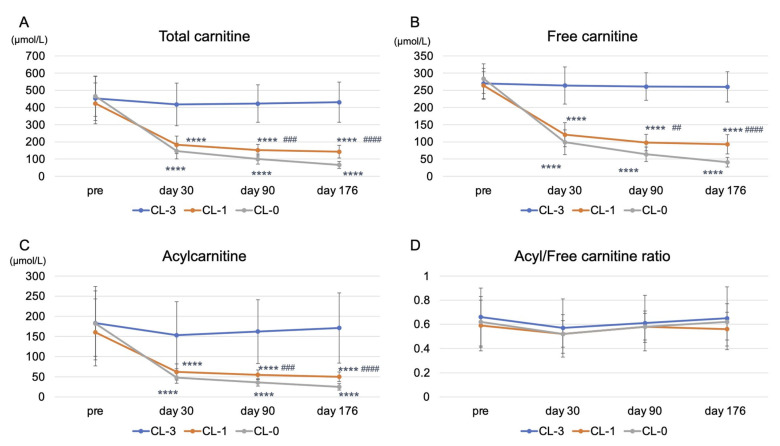
Kinetics of plasma total carnitine, free carnitine, acylcarnitine levels, and acyl/free carnitine ratio during the study. (**A**) plasma total carnitine levels; (**B**) plasma free carnitine levels; (**C**) plasma acylcarnitine levels; (**D**) acyl/free carnitine ratio. LC-3: LC 1000 mg three times weekly; LC-1: LC 1000 mg once and placebo (saline 5 mL) twice weekly; LC-0: placebo three times weekly (LC-0). **** *p* < 0.0001 vs. LC-3; ^##^
*p* < 0.01; ^###^
*p* < 0.001; ^####^
*p* < 0.0001 vs. LC-0.

**Figure 4 nutrients-13-01900-f004:**
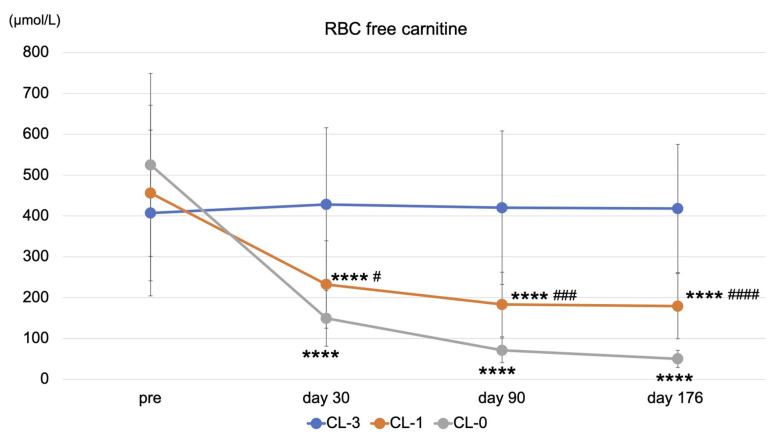
Kinetics of RBC-free carnitine level during the study. LC-3: LC 1000 mg three times weekly; LC-1: LC 1000 mg once and placebo (saline 5 mL) twice weekly; LC-0: placebo three times weekly (LC-0). **** *p* < 0.0001 vs. LC-3; ^#^
*p* < 0.05; ^###^
*p* < 0.001; ^####^
*p* < 0.0001 vs. LC-0.

**Figure 5 nutrients-13-01900-f005:**
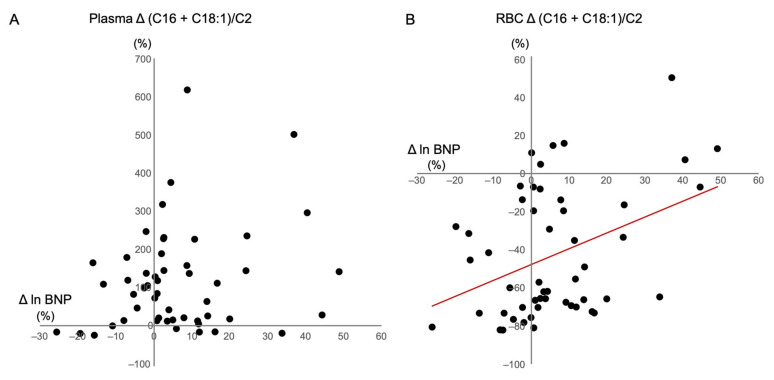
Correlation between changes in BNP with (C16 + C18:1)/C2 in (**A**) plasma; and (**B**) RBCs. BNP: B-type natriuretic peptide; RBC: red blood cell.

**Table 1 nutrients-13-01900-t001:** Clinical characteristics at baseline.

	LC 3 Times/w (LC-3)	LC 1 Time/w + Placebo 2 Times/w (LC-1)	Placebo 3 Times/w (LC-0)	*p* (LC-3 vs. LC-1)	*p* (LC-3 vs. LC-0)	*p* (LC-1 vs. LC-0)
Number	18	16	17	N/A		
Age (years)	64.5 ± 13.3	69.4 ± 12.7	69.5 ± 9.8	0.471	0.438	0.999
Sex (no.) (male/female)	12/6	10/6	10/7	N/A		
HD duration (months)	178 ± 91	150 ± 95	164 ± 120	0.721	0.916	0.926
Duration of LC treatment (months)	40 ± 21	35 ± 21	44 ± 27	0.788	0.860	0.484
Body weight (kg)	61.0 ± 13.2	57.4 ± 11.0	54.6 ± 10.7	0.656	0.249	0.765
Systolic BP (mmHg)	151 ± 27	141 ± 31	145 ± 24	0.516	0.792	0.892
Hemoglobin (g/dL)	11.2 ± 0.9	11.4 ± 0.7	11.7 ± 0.9	0.853	0.263	0.577
Total protein (g/dL)	6.26 ± 0.46	6.30 ± 0.51	6.45 ± 0.39	0.967	0.427	0.598
Serum albumin (g/dL)	3.57 ± 0.36	3.49 ± 0.27	3.54 ± 0.27	0.701	0.952	0.870
Total cholesterol (mg/dL)	164 ± 31	155 ± 28	159 ± 29	0.699	0.901	0.925
LDL-cholesterol (mg/dL)	89 ± 21	81 ± 20	88 ± 22	0.445	0.967	0.602
BUN (mg/dL)	56.6 ± 11.8	52.2 ± 14.0	61.5 ± 16.1	0.633	0.554	0.145
Corrected Ca (mg/dL)	8.73 ± 0.65	8.63 ± 0.39	8.49 ± 0.45	0.849	0.362	0.707
Phosphate (mg/dL)	4.64 ± 0.86	4.05 ± 0.90	4.97 ± 1.30	0.236	0.616	0.038
Plasma total carnitine (μmol/L)	454 ± 128	424 ± 119	466 ± 117	0.761	0.948	0.580
Plasma free carnitine (μmol/L)	270 ± 44	264 ± 40	284 ± 43	0.901	0.601	0.365
Plasma acylcarnitine (μmol/L)	183 ± 91	160 ± 83	182 ± 81	0.712	0.999	0.742
Acyl/free ratio	0.66 ± 0.24	0.59 ± 0.21	0.62 ± 0.21	0.633	0.909	0.872
RBC-free carnitine (μmol/L)	407 ± 203	456 ± 215	525 ± 224	0.781	0.242	0.629
LVEF (%)	65.3 ± 13.9	65.5 ± 10.1	65.8 ± 7.3	0.999	0.990	0.996
E/e’	13.9 ± 6.0	14.2 ± 7.1	13.2 ± 5.7	0.990	0.942	0.894
LVMI (g/m^2^)	122 ± 26	108 ± 23	115 ± 44	0.420	0.786	0.817
BNP ^†^ (pg/mL)	317 (34–1150)	424 (15–1380)	460 (44–1440)	0.866	0.541	0.858

Acyl/free: acylcarnitine/free carnitine; BNP: B-type natriuretic peptide; BUN: blood urea nitrogen; Ca: calcium; E/e’: ratio of early transmitral velocity E to mitral annular early diastolic velocity e′; HD: hemodialysis; LC: L-carnitine; LVMI: left ventricular mass index; LDL: low-density lipoprotein; RBC: red blood cell. ^†^ Because of the skewness of the BNP distribution, a natural logarithmic transformation was used for analysis.

**Table 2 nutrients-13-01900-t002:** Clinical characteristics at baseline and day 176 of the study period.

	LC 3 Times/w			LC 1 Time/w + Placebo 2 Times/w			Placebo 3 Times/w		
	Pre	Day 176	*p*	Pre	Day 176	*p*	Pre	Day 176	*p*
Body weight (kg)	61.0 ± 13.2	60.0 ± 13.0	0.036	57.4 ± 11.0	56.8 ± 10.4	0.063	54.6 ± 10.7	54.4 ± 11.2	0.555
Systolic BP (mmHg)	151 ± 27	154 ± 25	0.613	141 ± 31	149 ± 30	0.149	145 ± 24	140 ± 24	0.416
Hemoglobin (g/dL)	11.2 ± 0.9	11.0 ± 0.8	0.472	11.4 ± 0.7	11.2 ± 0.8	0.455	11.7 ± 0.9	11.4 ± 1.2	0.274
Total protein (g/dL)	6.26 ± 0.46	6.38 ± 0.43	0.043	6.30 ± 0.51	6.40 ± 0.40	0.377	6.45 ± 0.39	6.31 ± 0.40	0.079
Serum albumin (g/dL)	3.57 ± 0.36	3.67 ± 0.37	0.012	3.49 ± 0.27	3.50 ± 0.26	0.849	3.54 ± 0.27	3.50 ± 0.30	0.311
Total cholesterol (mg/dL)	164 ± 31	169 ± 30	0.044	155 ± 28	157 ± 24	0.765	159 ± 29	157 ± 30	0.573
LDL-cholesterol (mg/dL)	89 ± 21	96 ± 20	0.002	81 ± 20	83 ± 18	0.522	88 ± 22	87 ± 22	0.908
BUN (mg/dL)	56.6 ± 11.8	60.1 ± 12.6	0.052	52.2 ± 14.0	57.8 ± 16.5	0.065	61.5 ± 16.1	65.4 ± 19.0	0.211
Corrected Ca (mg/dL)	8.73 ± 0.65	8.78 ± 0.56	0.655	8.63 ± 0.39	8.73 ± 0.40	0.426	8.49 ± 0.45	8.48 ± 0.51	0.961
Phosphate (mg/dL)	4.64 ± 0.86	4.99 ± 0.76	0.180	4.05 ± 0.90	4.32 ± 1.21	0.373	4.97 ± 1.30	4.93 ± 1.36	0.910
Plasma total carnitine (μmol/L)	454 ± 128	431 ± 117	0.318	424 ± 119	143 ± 37	<0.0001	466 ± 117	66 ± 20	<0.0001
Plasma free carnitine (μmol/L)	270 ± 44	260 ± 44	0.383	264 ± 40	93 ± 28	<0.0001	284 ± 43	41 ± 14	<0.0001
Plasma acylcarnitine (μmol/L)	183 ± 91	171 ± 87	0.348	160 ± 83	50 ± 12	<0.0001	182 ± 81	25 ± 8	<0.0001
Acyl/free ratio	0.66 ± 0.24	0.65 ± 0.26	0.955	0.59 ± 0.21	0.56 ± 0.15	0.631	0.62 ± 0.21	0.62 ± 0.14	0.963
RBC-free carnitine (μmol/L)	407 ± 203	418 ± 157	0.717	456 ± 215	179 ± 80	<0.0001	525 ± 224	50 ± 21	<0.0001

Acyl/free: acylcarnitine/free carnitine; BP: blood pressure; BUN: blood urea nitrogen; Ca: calcium; LC: L-carnitine; LDL: low-density lipoprotein; RBC: red blood cell.

**Table 3 nutrients-13-01900-t003:** Effects of reducing or stopping LC treatment on LV function, LVMI, and BNP levels.

	LC 3 Times/w			LC 1 Time/w + Placebo 2 Times/w			Placebo 3 Times/w		
	Pre	Day 176	*p*	Pre	Day 176	*p*	Pre	Day 176	*p*
LVEF (%)	65.3 ± 13.9	63.7 ± 10.8	0.446	65.5 ± 10.1	65.4 ± 7.6	0.967	65.8 ± 7.3	65.3 ± 7.4	0.774
E/e’	13.9 ± 6.0	14.1 ± 7.0	0.737	14.2 ± 7.1	14.0 ± 6.2	0.893	13.2 ± 5.7	15.1 ± 8.9	0.224
LVMI (g/m^2^)	122 ± 26	129 ± 33	0.330	108 ± 23	107 ± 39	0.873	115 ± 44	120 ± 44	0.207
BNP ^†^ (pg/mL)	317 (34–1150)	396 (51–1270)	0.255	424 (15–1380)	433 (57–1580)	0.422	460 (44–1440)	631 (44–1450)	0.030

BNP: B-type natriuretic peptide; E/e’: ratio of early transmitral velocity E to mitral annular early diastolic velocity e′; LC: L-carnitine; LV: left ventricular; LVMI: left ventricular mass index. ^†^ Because of the skewness of the BNP distribution, a natural logarithmic transformation was used for analysis.

**Table 4 nutrients-13-01900-t004:** Univariate and multiple regression analyses for the covariates of changes in BNP.

	Univariate Regression			Multiple Regression		
	β	SE	*p*	β	SE	*p*
∆Body weight	0.069	0.024	0.630			
∆Systolic blood pressure	0.373	0.152	0.007	0.331	0.109	0.011
∆Hemoglobin	−0.077	0.097	0.590			
∆Total protein	−0.038	0.051	0.789			
∆Albumin	−0.032	0.052	0.826			
∆Total cholesterol	0.000	0.095	0.999			
∆LDL-cholesterol	−0.106	0.145	0.461			
∆Blood urea nitrogen	−0.220	0.180	0.121			
∆Corrected Ca	0.011	0.051	0.441			
∆Phosphate	−0.144	0.285	0.312			
∆Plasma total carnitine	−0.114	0.349	0.427			
∆Plasma free carnitine	−0.135	0.344	0.345			
∆Plasma acylcarnitine	−0.063	0.352	0.661			
∆Acyl/Free ratio	0.229	0.233	0.110			
∆Plasma (C16 + C18:1)/C2	0.219	1.191	0.123			
∆Plasma C0/(C16 + C18)	−0.156	1458.7	0.273			
∆Plasma C8/C10	−0.206	0.160	0.147			
∆Plasma C14/C3	0.016	2.451	0.912			
∆RBC (C16 + C18:1)/C2	0.389	0.283	0.005	0.350	0.058	0.007
∆RBC C0/(C16 + C18)	−0.131	0.282	0.359			
∆RBC C8/C10	−0.154	0.218	0.281			
∆RBC C14/C3	0.160	0.923	0.160			

R^2^ = 0.259. Acyl/free: acylcarnitine/free carnitine; BNP: B-type natriuretic peptide; Ca: calcium; ∆: delta; LDL: low-density lipoprotein; RBC: red blood cell.

## Data Availability

The data presented in this study are available on request from the corresponding author.

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
