# Peer review of "Effects of Reducing L-Carnitine Supplementation on Carnitine Kinetics and Cardiac Function in Hemodialysis Patients: A Multicenter, Single-Blind, Placebo-Controlled, Randomized Clinical Trial"

_nutrients, 2021, doi:10.3390/nu13061900_

Round 1

Reviewer 1 Report

In this randomized control trial (RCT), Dr. Sugiyama and colleagues studied the effects of the reduction and termination of L-carnitine (LC) supplementation in end-stage renal disease (ESRD) patients undergoing hemodialysis (HD) on carnitine kinetics and cardiac function. They nicely demonstrated that after 6 months, reducing LC therapy significantly decreased both plasma and blood carnitine levels, and terminating LC increased plasma BNP levels. Overall, the RCT was conducted appropriately and the manuscript is well-written. I have some suggestions to improve the readability of the manuscript and some questions to be addressed by the authors:

  • I think the authors missed one more observation group to be studied. I am curious what the effect will be if the patients obtain a reduction of LC dose from 1000 3/w to 500 3/w for 6 months, instead of 1000 1/w + saline 2/w? Can the authors speculate on this?
  • Table 1 has to be divided into 2 tables. The baseline characteristics should be in a separate table without the results on day-176. I would suggest to move the results of day-176 to a new table. Also, please assess the parameter difference between groups at baseline, whether they are statistically significant or not.
  • Because Table 1 is too complex, the data confirming this sentence "The baseline phosphate level was higher in the LC-0 group than in the LC-1 group (p < .05) (Table 1)" is not visible anymore. Moreover, it is currently not clear which comparison is represented by the p-value (baseline vs. day-176 or comparison between groups at baseline)?
  • Please also add the baseline cardiac function in Table 1 and assess the statistical difference to evaluate whether there was a notable baseline difference between groups. 
  • The font sizes in Figure 3 and 4 are too small and the figure presentation can still be improved. Consider revising. 
  • Please provide figure captions for Figure 1 and 2.
  • Please add some recommendations for future research after the limitation of the study. This would be a useful addition to the manuscript. 
  • Please change the keywords to improve the visibility of the work. They are too long and general.
  • "Heart failure (HF) is a serious complication characterized ...". Complication of what? Please clarify.
  • "...abnormal hemodynamic states in patients..." should be "status" or "state".
  • Please revise this sentence in the abstract: "Fifty-nine HD patients previously treated with intravenous LC 1000 mg per HD session (three times a week) were allocated to three groups ..."
  • Please revise this: "Reducing LC therapy for 6 months significantly decreased both plasma and RBC carnitine levels, while the full termination of LC increased plasma BNP levels. However, it did not influence cardiac function in HD patients."
  • In this sentence "Patients over 20 years of age with end-stage renal disease undergoing HD who were able...", ESKD has been introduced before, so please change to ESKD or ESRD, whichever preferred. Please also check for similar errors. Basically, once introduced, please always use abbreviation throughout the manuscript.

Author Response

  • In this randomized control trial (RCT), Dr. Sugiyama and colleagues studied the effects of the reduction and termination of L-carnitine (LC) supplementation in end-stage renal disease (ESRD) patients undergoing hemodialysis (HD) on carnitine kinetics and cardiac function. They nicely demonstrated that after 6 months, reducing LC therapy significantly decreased both plasma and blood carnitine levels, and terminating LC increased plasma BNP levels. Overall, the RCT was conducted appropriately and the manuscript is well-written. I have some suggestions to improve the readability of the manuscript and some questions to be addressed by the authors:

Ans. Thank you very much for your positive comments.

  • I think the authors missed one more observation group to be studied. I am curious what the effect will be if the patients obtain a reduction of LC dose from 1000 3/w to 500 3/w for 6 months, instead of 1000 1/w + saline 2/w? Can the authors speculate on this?

Ans. Thank you for your valuable comments. As you mentioned, the effects of reducing LC dose (from 1000 mg 3/w to 500 mg 3/w) on cardiac function might be interesting. However, because one vial contains 1000 mg of LC, administering 500 mg of this will render the remaining 500 mg wasted. Further, this reduction (1000 mg 1/w) is common in the clinical setting. I speculate that reducing to 500 mg 3/w does not influence BNP levels and cardiac function because our preliminary data showed that LC reduction to 500 mg increased plasma LC concentration to approximately 150 μmol/L, which is still higher than that in patients receiving LC 1000 mg 1/w.

  • Table 1 has to be divided into 2 tables. The baseline characteristics should be in a separate table without the results on day-176. I would suggest to move the results of day-176 to a new table. Also, please assess the parameter difference between groups at baseline, whether they are statistically significant or not.

Ans. Thank you for your comments.  Table 1 has been revised, excluding the results on day-176. Further, I made a new Table 2 including clinical data of day-176. We analyzed the parameter difference between groups at baseline, and only the baseline phosphate level was higher in the LC-0 group than in the LC-1 group (p < .05). Accordingly, we have provided below sentence in the Results section of the revised manuscript.

No significant differences in age, HD duration, duration of previous LC treatment, or other clinical parameters, except for phosphate, were observed among the groups (data not shown). The baseline phosphate level was higher in the LC-0 group than in the LC-1 group (p < .05) (Table 1). (Page 5)

  • Because Table 1 is too complex, the data confirming this sentence "The baseline phosphate level was higher in the LC-0 group than in the LC-1 group (p < .05) (Table 1)" is not visible anymore. Moreover, it is currently not clear which comparison is represented by the p-value (baseline vs. day-176 or comparison between groups at baseline)?

Ans. Thank you for your comments. Tables 1 and 2 were revised for easier understanding. In Table 2, the comparison of clinical variables and carnitine fractions between baseline and day 176 of the study period was represented by the p-value. Accordingly, we have added this sentence to the Results section of the revised manuscript. (Page 6)

  • Please also add the baseline cardiac function in Table 1 and assess the statistical difference to evaluate whether there was a notable baseline difference between groups.

Ans. Thank you for your advice. Descriptions regarding the baseline cardiac function were moved to Table 1. The baseline cardiac function and BNP levels were not different among the study groups (data not shown). Accordingly, we have added this sentence to the Results section of the revised manuscript. (Page 5)

  • The font sizes in Figure 3 and 4 are too small and the figure presentation can still be improved. Consider revising.

Ans. Thank you for your comments. Accordingly, the font size in figures 3 and 4 has been improved.

  • Please provide figure captions for Figure 1 and 2.

Ans. Thank you for your comments. Accordingly, captions have been provided for figures 1 and 2.

  • Please add some recommendations for future research after the limitation of the study. This would be a useful addition to the manuscript.

Ans. Thank you for your comments. Carnitine deficiency induces mitochondrial damages through the impairment of β-oxidation and ATP production. However, the exact mechanism underlying the exac-erbation of cardiomyocyte injury by hemodialysis-induced carnitine deficiency through the manipulation of carnitine-related mitochondrial enzymes, such as CPT1 and CPT2, is not fully understood. Hence, future research using a carnitine-deficient experimental model might be required to examine the mediating mechanism underlying carnitine deficiency-associated myocardial injury.

Accordingly, we have added these sentences to the Discussion section of the revised manuscript. (Page 11)

  • Please change the keywords to improve the visibility of the work. They are too long and general.

Ans. Thank you for your suggestion. Accordingly, we have revised the keywords to improve visibility as follows.

acylcarnitine; brain natriuretic peptide; cardiac function; cardiomyopathy; carnitine deficiency; CPT2; end-stage kidney disease; free fatty acid; heart failure; hemodialysis.

  • "Heart failure (HF) is a serious complication characterized ...". Complication of what? Please clarify."...abnormal hemodynamic states in patients..." should be "status" or "state".

Thank you for your comments. We have revised the sentence as follows:

Heart failure (HF), as well as some of its complications, such as pulmonary edema, is a serious condition characterized by decreased myocardial contractility and abnormal hemodynamic state in patients with end-stage kidney disease (ESKD). (Page 1)

  • Please revise this sentence in the abstract: "Fifty-nine HD patients previously treated with intravenous LC 1000 mg per HD session (three times a week) were allocated to three groups ..."

Ans. Thank you for your comments. The sentence has been revised accordingly.

  • Please revise this: "Reducing LC therapy for 6 months significantly decreased both plasma and RBC carnitine levels, while the full termination of LC increased plasma BNP levels. However, it did not influence cardiac function in HD patients."

Ans. Thank you for your comments. The sentence has been revised accordingly.

  • In this sentence "Patients over 20 years of age with end-stage renal disease undergoing HD who were able...", ESKD has been introduced before, so please change to ESKD or ESRD, whichever preferred. Please also check for similar errors. Basically, once introduced, please always use abbreviation throughout the manuscript.

Ans. Thank you for your comments. Accordingly, I have checked the whole manuscript for errors regarding abbreviations.

Reviewer 2 Report

The article entitled “Effects of Reducing L-Carnitine Supplementation on Carnitine Kinetics and Cardiac Function in Hemodialysis Patients: A Multicenter, Single-Blind, Placebo-Controlled, Randomized Clinical Trial” is a current and interesting topic in hemodialysis patients. Some comments of the manuscript are suggested to the authors.

 As a RCT, diagram should be adapted to the Consort diagram. Figure 2, is before that figure 1.

Please, I would like to know what kind of haemodialysis technique was used?

Was the study registered as an RCT and evaluated by an Ethics Committee?

Were there carnitine-deficient patients at the start of the study?.Please, explain it

Did you find any adverse effects in this study?

From a statistical perspective, please explain me why do you use univariate and multivariate regression analyses in this study?. The effects of the L- carnitine supplementation are not sufficiently demonstrated. How were the main effects of L-carnitine assessed post-intervention compared to baseline between groups?. Only differences between the same intervention group are provided. Please, explain it

Author Response

  • The article entitled “Effects of Reducing L-Carnitine Supplementation on Carnitine Kinetics and Cardiac Function in Hemodialysis Patients: A Multicenter, Single-Blind, Placebo-Controlled, Randomized Clinical Trial” is a current and interesting topic in hemodialysis patients. Some comments of the manuscript are suggested to the authors.

Thank you for your positive comments.

  • As a RCT, diagram should be adapted to the Consort diagram. Figure 2, is before that figure 1.

Ans. Thank you for your comments. The illustration has been adapted to the CONSORT diagram, and figure 2 has been moved before figure 1.

  • Please, I would like to know what kind of haemodialysis technique was used?

Ans. Thank you for your comments. We performed hemodialysis for 4–5 h with high-flux dialyzers three times weekly, as already mentioned in the manuscript. We neither performed hemodiafiltration nor continuous hemofiltration.

  • Was the study registered as an RCT and evaluated by an Ethics Committee?

Ans. Thank you for your comments. This study was conducted according to the guidelines of the Declaration of Helsinki and registered with the University Hospital Medical Information Network clinical trials database (UMIN000037330). The study protocol was approved by the Institutional Ethics Committees of Kurume University School of Medicine (Approval Number: 18179). Accordingly, we have provided the sentences above in the manuscript as the Institutional Review Board Statement. (Page 13)

  • Were there carnitine-deficient patients at the start of the study? Please, explain it

Thank you for your comments. All patients had already been diagnosed with dialysis-associated secondary carnitine deficiency and administered intravenous LC at a dose of 1000 mg per HD session three times weekly for at least three months. Accordingly, this sentence has been added to the Materials and methods section of the revised manuscript. (Page 2)

  • Did you find any adverse effects in this study?

Ans. Thank you for your valuable comments. No significant adverse events related to LC treatment stoppage were observed during the study period. Accordingly, we have added this statement to the Results section of the revised manuscript. (Page 5)

  • From a statistical perspective, please explain me why do you use univariate and multivariate regression analyses in this study?. The effects of the L- carnitine supplementation are not sufficiently demonstrated. How were the main effects of L-carnitine assessed post-intervention compared to baseline between groups?. Only differences between the same intervention group are provided. Please, explain it

Ans. Thank you for your valuable question. We observed that the cessation of LC, rather than its reduction, significantly increased BNP levels before and after the study period in HD patients. This is the main effect of stopping LC treatment assessed post-intervention compared to baseline. As a next step, To further explore which valuables could be independently associated with BNP levels, we assessed correlations between changes in BNP and clinical variables and acylcarnitine ratios, such as (C16 + C18:1)/C2, a marker of CPT2 deficiency; C0/(C16 + C18), a marker of CPT1 deficiency; C8/C10, a marker of medium-chain acyl-CoA dehydrogenase  deficiency; and C14/C3, another marker of CPT2 deficiency, by univariate and multiple regression analyses in all HD patients. We speculated that independent determinants of the changes in BNP might be involved in the pathogenesis of carnitine deficiency-associated HF.

We have revised the sentences and provided in the Results section. (Page 9)

Round 2

Reviewer 1 Report

Thank you for addressing my questions/comments. I only have one minor suggestion:

  • please add the p-values next to each line in Table 1. At the moment, we don't know which one is statistically significant.

Reviewer 2 Report

Dear authors, 

 Thanks for your answers  and for the changes made to the manuscript. This topic is interesting and provide  valuable information from a clinical perspective in hemodialysis patients